# Tuning Strain Stiffening of Protein Hydrogels by Charge Modification

**DOI:** 10.3390/ijms23063032

**Published:** 2022-03-11

**Authors:** Jie Gu, Yu Guo, Yiran Li, Juan Wang, Wei Wang, Yi Cao, Bin Xue

**Affiliations:** 1Collaborative Innovation Center of Advanced Microstructures, National Laboratory of Solid State Microstructure, Key Laboratory of Intelligent Optical Sensing and Manipulation, Ministry of Education, Department of Physics, Nanjing University, Nanjing 210093, China; 131242017@smail.nju.edu.cn (J.G.); liyiran0628@nju.edu.cn (Y.L.); wangjuannm@163.com (J.W.); wangwei@nju.edu.cn (W.W.); caoyi@nju.edu.cn (Y.C.); 2College of Life and Health Sciences, Northeastern University, Shenyang 110819, China; 1901405@stu.neu.edu.cn

**Keywords:** protein hydrogel, strain-stiffening, electrical repulsion, surface charge, mechanical property

## Abstract

Strain-stiffening properties derived from biological tissue have been widely observed in biological hydrogels and are essential in mimicking natural tissues. Although strain-stiffening has been studied in various protein-based hydrogels, effective approaches for tuning the strain-stiffening properties of protein hydrogels have rarely been explored. Here, we demonstrated a new method to tune the strain-stiffening amplitudes of protein hydrogels. By adjusting the surface charge of proteins inside the hydrogel using negatively/positively charged molecules, the strain-stiffening amplitudes could be quantitively regulated. The strain-stiffening of the protein hydrogels could even be enhanced 5-fold under high deformations, while the bulk property, recovery ability and biocompatibility remained almost unchanged. The tuning of strain-stiffening amplitudes using different molecules or in different protein hydrogels was further proved to be feasible. We anticipate that surface charge adjustment of proteins in hydrogels represents a general principle to tune the strain-stiffening property and can find wide applications in regulating the mechanical behaviors of protein-based hydrogels.

## 1. Introduction

Strain-stiffening behaviors are normally observed in natural soft tissues and are defined as an increase in stiffness under applied strains, thus protecting the system from large deformations [1,2]. Strain-stiffening is also expected to be essential for cell differentiation [3]. This unique characteristic is usually derived from cytoskeletal structures formed by proteins, including actin [4], collagen [5,6], fibrin [7,8], and other types of intermediate filaments [9]. Inspired by the structure of biological tissues, synthetic materials exhibiting strain-stiffening behaviors, such as hydrogels formed by biopolymers or the self-assembly of biomolecules, have been developed by researchers [1,10,11,12,13]. Recently, hydrogels formed with polysaccharides such as alginate [14], methylcellulose [15], and pectin [16,17] were found to exhibit strain-stiffening behaviors. In addition to the above hydrogels, hydrogels with proteins acting as crosslinkers can also show strain-stiffening due to the electrical repulsion between the charged biomacromolecules. Due to their programmable and tunable properties, protein hydrogels have been widely studied for various biological applications, including tissue engineering [18,19,20], biosensors [21,22], drug delivery [23], and wound dressings [24,25]. Although various efforts have been made to regulate the mechanical properties of protein hydrogels, effective methods to tune strain-stiffening behaviors have rarely been explored [26,27,28,29,30].

Moment by moment, our articular cartilage bears the great load brought by daily exercise. Articular cartilage is a matrix exquisitely weaved by collagen and proteoglycans and embedded with chondrocytes, which would especially benefit strain-stiffening behaviors under deformations [31]. Due to its high stiffness (0.24–0.85 MPa) and large energy dissipation and strain-stiffening properties, cartilage can hold several times its body weight and quickly recover under cyclic-stress loading (frequency over 1 Hz, peak pressures ranging from 1–20 MPa) [32]. In articular cartilage, the aggregating proteoglycans entrapped within the collagen matrix are also critical to biological functions. These highly charged proteoglycans attract cations and water, thus increasing osmolality [13]. Moreover, highly charged proteoglycans inside the cartilage repel each other under large deformations, thus contributing to the strain-stiffening ability (Figure 1A).

Inspired by the electrorepulsive interaction in highly stress-tolerant cartilage, we demonstrated a new method to tune the strain-stiffening properties of protein-based hydrogels without affecting hydrogel networks. By adjusting the surface charge of BSA using charged molecules, the strain-stiffening could be tuned while the bulk property, microstructure and recovery ability were barely affected. Furthermore, this method of tuning strain-stiffening was demonstrated to be feasible using different molecules or applicable for different protein hydrogels. Finally, the outstanding biocompatibility of protein hydrogels was also proved to be unchanged. We expect that the surface charge modification of proteins in hydrogels can be widely applied to tune the strain-stiffening behaviors without affecting the bulk properties and biocompatibilities.

## 2. Results and Discussion

### 2.1. Design of Tuning the Strain-Stiffening of Protein Hydrogels by Protein Surface Charge Modification

As shown in Figure 1B, BSA-based hydrogels were chosen as the model protein hydrogel due to their stable mechanical properties and excellent biocompatibility [33,34,35,36,37]. By mixing BSA protein solutions with N-hydroxysuccinimide-terminated 4-armed PEG (Mw: 20 kDa, named PEG), transparent hydrogels (denoted as BSA-PEG hydrogels) formed immediately in several minutes. The BSA protein acts as the crosslinker, while PEG acts as the soft link between proteins. The average distance between the negatively charged BSA molecules would decrease with the deformation of hydrogels, leading to an increase in electrostatic repulsion. The mechanical resistance of the hydrogels under large strains would also increase as a result of the enhanced electrostatic interactions (Figure 1B). Furthermore, the negative surface charge of BSA can be enhanced by connecting the negatively charged molecule to the amino groups on the surface of BSA (Figure 1C). We expect that the strain-stiffening of the hydrogel built with the modified BSA under large deformations to be significantly increased due to the increased surface charge of BSA. On the other hand, the surface charge of the protein can also be tuned down through the modification of oppositely charged molecules, and the resulting hydrogels might exhibit weaker mechanical resistance under large strains. Because the Debye length is small in hydrogels, the effects of the surface charge of BSA on the strain-stiffening of hydrogels are supposed to be observed under large deformations. As a result, the bulk properties and mechanical properties at low strains would be nearly unaffected. Moreover, the electrical repulsion between trapped charges is correlated with ${\left(nq\right)}^{2}$ while that between the dispersive charges is correlated with $nq^{2}$, in which q corresponds to the single charge value and n corresponds to the number of single charges. We anticipate that globular proteins with surface charges trapped as an entirety are supposed to most likely enhance the strain-stiffening property of hydrogels. We expected that such modification of the protein surface charge could be used to tune the strain-stiffening of protein hydrogels without affecting the bulk properties and microstructures.

### 2.2. Mechanical Properties of BSA-PEG Hydrogels

The mechanical properties of BSA-PEG hydrogels without modification were investigated first. As shown in Figure 2A, the compressive mechanical properties were estimated by pressing the hydrogel directly. The compressibility of hydrogels with different ratios and concentrations of BSA and PEG was studied to explore the effects on the mechanical properties (Figure 2B). The Young’s modulus of the BSA-PEG hydrogels increased from 20 kPa to 50 kPa, while the toughness varied in the range of ~38–56 kJ m^−3^. Obviously, the hydrogel exhibited relatively higher mechanical strength at BSA and PEG concentrations of 10 mM and 5 mM, indicating the optimized hydrogel network (Figure 2C,D). Furthermore, the strain-stiffening behavior of the hydrogels was also studied. The differential modulus of BSA-PEG hydrogels increased obviously with increasing strain, exhibiting strain-stiffening properties (Figure 2E). Moreover, the BSA-PEG hydrogels exhibited obvious energy dissipation, which was estimated by applying loading–unloading cycles at different strains (Appendix A). The recovery property of the BSA-PEG hydrogel was also measured by applying 10 loading–unloading cycles to the same hydrogel continuously (Appendix A). The maximum stress and dissipated energy of unmodified BSA-PEG hydrogels remained ~85% and ~70% of the original values after 10 cycles of compression–relaxation, indicating a certain recovery ability (Appendix A). These results suggested that BSA-PEG hydrogels exhibited reliable mechanical performance similar to commonly manufactured protein hydrogels [22,27,38,39]. BSA and PEG concentrations of 10 mM and 5 mM were used to prepare the hydrogels as a model for the regulation of the strain-stiffening property thereafter.

### 2.3. Charge Modification and Bulk Properties of the BSA-PEG Hydrogels

For the surface charge modification of BSA, the protein was dissolved into glyoxylic acid solutions to replace the amino groups on the BSA surface with carboxyl groups. By adjusting the ratios of glyoxylic acid and BSA, the surface charge of BSA can be tuned to different extents. As indicated by the zeta potential, the absolute value of the surface charge of BSA first increased from ~9.3 mV to ~33.0 mV, and then decreased to ~21.0 mV with increasing glyoxylic acid:BSA ratio (0:0, 1:1, 2:1, 2.5:1, 3:1 and 4:1, simplified as 0, 1.0, 2.0, 2.5, 3.0 and 4.0). However, the reason for the decrease of zeta potential at high glyoxylic acid concentrations remains unknown, and needs to investigated in the future.

To investigate the effects of surface charge modification on the structure of BSA, modified BSA was studied using circular dichroism (CD) spectroscopy and thiol detection. As shown in Figure 3B, the CD spectra of BSA modified with different ratios of glyoxylic acid and BSA were almost the same. The surface charge modification of BSA in hydrogels was achieved by immersing the hydrogel into glyoxylic acid solutions, similar to the modification of BSA. The amounts of free thiol in BSA-PEG hydrogels were detected with 5,5′-dithiobis-(2-nitrobenzoic acid) (DTNB) to evaluate the unfolding of BSA in hydrogels after modification using glyoxylic acid. The color of the DTNB solutions containing modified hydrogels and the UV absorbance at 412 nm were also almost the same at different glyoxylic acid:BSA ratios, suggesting the mild unfolding of BSA in the hydrogel (Figure 3C and Appendix A). The modified BSA-PEG hydrogels were split into particles to expose the inner surfaces and the zeta potentials were measured. As shown in Appendix A, the samples exhibited zeta potentials and pH values similar to those of the modified BSA, suggesting the successful surface charge modification of BSA inside hydrogels. Furthermore, the bulk properties of the modified hydrogel were studied in detail. The swelling ratios remained the same after the modification of glyoxylic acid, and the porosity was also kept at 91%, indicating that the charge modification would not affect the bulk properties of hydrogels (Appendix A). Then, the microstructure of the modified BSA-PEG hydrogels was studied using scanning electron microscopy (SEM) (Figure 3D). The pore sizes of different hydrogels are summarized in Appendix A. Obviously, all the hydrogels exhibited similar pore sizes, suggesting mild effects of the modification on microstructures. All the results demonstrated that the modification of surface charge would mildly affect the folding structure of BSA or the bulk properties and microstructures of BSA-PEG hydrogels.

### 2.4. Tuning Strain-Stiffening of BSA-PEG Hydrogels by Charge Modification

Next, the mechanical properties of BSA-PEG hydrogels with BSA surface charge modified by glyoxylic acid were studied in detail by compression tests. The typical stress–strain curves corresponding to modified and unmodified hydrogels almost overlapped in the region of low strains (<10%), indicating the mild effects of BSA modification on hydrogels under small deformations (Figure 4A). The negligible difference under small deformations was further confirmed by the similar Young’s modulus of all the hydrogels (Appendix A, calculation range: 0–10%). In contrast, the stress of the modified hydrogels grows higher than that of the unmodified hydrogels with increasing strain, probably due to the increased electrical repulsion of modified BSA. The differential modulus of modified BSA-PEG hydrogels at different strains is summarized in Figure 4B. The differential modulus of all the hydrogels increased with increasing strain scale, exhibiting strain-stiffening behaviors. The increase amplitude of the differential modulus for modified hydrogels reached 2000–3300% in the strain range of 0–65% compared to 1100% of unmodified hydrogels, indicating that the strain stiffening behaviors were significantly enhanced by the surface charge modification of BSA. Please note that the fracture strains of the modified hydrogels were slightly decreased compared to those of the unmodified hydrogels, probably due to the higher internal stress under large deformations caused by the strong electrical repulsion of modified BSA.

The stress–strain curves for the compression–relaxation cycle at the strain of 50% for hydrogels modified with different ratios of glyoxylic acid and BSA are shown in Figure 4C. The maximum stress of the hydrogels at 50% showed similar trends with that of the absolute value of the zeta potential, further proving that the enhancement of mechanical strength was attributed to the surface charge modification of BSA (Figure 4D). Obvious hysteresis between the loading and unloading curves was observed for all the hydrogels due to the energy dissipation caused by protein unfolding [27,30,40]. Interestingly, the energy dissipation also exhibited the same trends as that of the absolute values of zeta potentials, probably due to the fact that strong electrical repulsion makes it easier to unfold the BSA proteins (Appendix A). It is worth mentioning that the stress at 50% of the unmodified hydrogels also showed a trend similar to that of the absolute values of zeta potentials for BSA at various pH (Appendix A), further confirming the effects of surface charge of BSA on the strain-stiffening of hydrogels. In addition to pH, the salt concentrations of solutions also affect the strain-stiffening of hydrogels. As shown in Appendix A, the strain-stiffening amplitude of BSA-PEG hydrogels in salted solutions decreased with increasing salt concentrations. The decreased Debye length in salted solutions leads to smaller repulsion between nearby BSA, resulting in decreased strain-stiffening amplitudes. It is worth mentioning that the contribution of intermolecular crosslinking of BSA to the strain-stiffening cannot be ruled out according to the SDS-PAGE gels of the modified BSA (Appendix A).

The mechanical properties of hydrogels prepared using the modified BSA were also evaluated for comparison, and similar results to those of the hydrogels modified after the gelation were observed (Appendix A). The increase amplitudes of the differential modulus first increased and then decreased with the molar ratios of glyoxylic acid and BSA (Appendix A). The maximum stress at the strain of 50% for the modified hydrogels also showed trends similar to those of the absolute value of the zeta potential (Appendix A). Interestingly, the differential modulus and intrinsic mechanical strength of the hydrogels prepared using modified BSA were lower than those of the unmodified hydrogels (Appendix A). Since the hydrogel was formed via the reaction between surface amino groups of BSA and NHS-terminated PEG, the modification of surface amino groups may lead to a decrease in the crosslinking densities.

Furthermore, the recovery properties of the hydrogels were investigated by applying compression and relaxation cycles to the same hydrogel without any waiting time (Figure 4E). The normalized maximum stress of different hydrogels in 10 compression–relaxation cycles is summarized in Figure 4F. The recovery percentage of all hydrogels modified at different glyoxylic acid:BSA ratios remained more than 80% after 10 cycles of compression and relaxation, suggesting that the recovery property of hydrogels was not affected by the surface charge modification of BSA. All these results demonstrated the important role of electrostatic repulsion interactions in the compressive performance of protein hydrogels. Charge modification can be used to tune the strain-stiffening property of protein hydrogels with mild effects on the energy dissipation and recovery properties.

### 2.5. Universality of Tuning Strain-Stiffening of Hydrogels by Surface Charge Modification of Proteins

To demonstrate the universality of tuning the strain-stiffening property of protein hydrogels by charge modification, BSA-PEG hydrogels modified using bromoacetic acid and hemoglobin-PEG hydrogels modified with glyoxylic acid were also studied. For BSA-PEG hydrogels modified with bromoacetic acid, the strain-stiffening property was also significantly enhanced, with the porosity, swelling ratios and pore size remaining unchanged (Figure 5A–C, Appendix A). The differential modulus of the modified hydrogel at a bromoacetic acid:BSA ratio of 0.6 increased by more than 1300% in the strain range of 0–55% compared to 675% of unmodified hydrogels (Figure 5B). The maximum stress of the modified hydrogels at the strain of 50% also showed a trend similar to that of the absolute value of zeta potentials, while the Young’s modulus and recovery properties remained slightly changed (Figure 5C and Appendix A).

Moreover, the mechanical properties of the hemoglobin-PEG hydrogels modified with glyoxylic acid are shown in Figure 5D–F. Since hemoglobin is positively charged under neutral conditions, the absolute value of the surface charge for hemoglobin was reduced by the modification of glyoxylic acid. The strain-stiffening property was also weakened with the porosity, swelling ratios and pore size remaining unchanged (Figure 5D–F and Appendix A). The increasing amplitude of the differential modulus of the modified hydrogel decreased to less than 342% in the strain range of 0–60%, while that of the unmodified hemoglobin-PEG hydrogel was 1270% (Figure 5D,E). The maximum stress of the modified hydrogels at the strain of 50% decreased with decreasing of the absolute values of zeta potentials while the Young’s modulus slightly changed (Figure 5F and Appendix A). Please note that the pH of all the protein solutions after the surface charge modification remained almost the same (~5.6), indicating that the enhancement of strain-stiffening properties was not due to the change in pH (Appendix A). All these results demonstrated that the surface charge modification of proteins in hydrogels can be a universal method to tune the strain-stiffening performance of protein hydrogels.

### 2.6. Biocompatibility of the Modified BSA-PEG Hydrogels

Finally, the cytotoxicity of BSA-PEG hydrogels modified with glyoxylic acid was studied, since protein-based hydrogels are widely used in cell culture, drug release and tissue engineering. Human high-metastatic liver cancer cells 97H (MHCC-97H) and kidney-2 cells (HK-2) were chosen to be cultured on hydrogels modified at different ratios of glyoxylic acid and BSA. After being cultured for 48 h, the cells were stained using calcein-AM and propidium iodide (PI). As shown in Appendix A, the cell morphologies of both 97H and HK-2 cells on modified hydrogels (Modified) were the same as those of cells cultured on unmodified hydrogels (Unmodified) or cell culture plates (Control). Living cells with high enzymatic activity spread throughout the hydrogels, indicating that cell spreading was also not affected by modification using glyoxylic acid. Almost no dead cells (stained in red) were found on the hydrogels. The viabilities of cells living on hydrogels modified with different ratios of BSA and glyoxylic acid were determined with Promega CellTiter-Glo (Appendix A). All the cell viabilities remained higher than 85%, suggesting the excellent biocompatibility of all the modified hydrogels.

## 3. Materials and Methods

Materials: BSA and hemoglobin were purchased from Shanghai Aladdin Biochemical Technology Co., Ltd. (Shanghai, China). Succinimidyl glutamate-terminated 4-armed polyethylene glycol (M.W. = 20 kDa) (4-armed PEG-SG) was purchased from Sinopeg Co., Ltd. (Xiamen, Fujian, China). Glyoxylic acid, bromoacetic acid, and DTNB were purchased from Shanghai Sigma–Aldrich Co., Ltd. (Shanghai, China). ddH_2_O was produced by a Milli-Q^®^ integral water purification system (Merck KGaA, Darmstadt, Germany). All other chemical reagents, unless otherwise stated, were purchased from Shanghai Aladdin Biochemical Technology Co., Ltd. (Shanghai, China). All reagents were used without further purification.

Surface charge modification of proteins: For the modification of BSA using glyoxylic acid, glyoxylic acid was dissolved in BSA solutions to different concentrations (0–3.6 mM). Then, the solutions were stirred for more than 24 h at room temperature. The mixtures were dialyzed in ddH_2_O for 48 h to remove the unreacted molecules and lyophilized. The modification of BSA using bromoacetic acid and the modification of hemoglobin using glyoxylic acid were achieved using the same methods.

Measurement of zeta potentials: Different proteins were dissolved in ddH_2_O to 15 μM, and then the suspension solutions were measured using a Zetasizer Nano ZS (Malvern, UK). The same protein solutions were measured at least three times to guarantee reproducibility.

UV-Vis spectroscopy: UV-Vis spectra of all samples were recorded using a V-550 (JASCO Inc., Tokyo, Japan) spectrophotometer. The cuvette width was 1 cm, and the bandwidth was 0.2 nm.

Preparation and charge modification of the BSA-PEG and hemoglobin-PEG hydrogels: four-armed PEG-SG and BSA were dissolved in ddH_2_O to concentrations of 20 mM and 10 mM, respectively. Then, the two kinds of solutions were quickly mixed at a volume ratio of 1:1. Transparent BSA-PEG hydrogels formed minutes after mixing. Then, the hydrogels were dialyzed in ddH_2_O for 24 h to remove the unreacted BSA and PEG. For the charge modification of the BSA-PEG hydrogels, BSA-PEG hydrogels were prepared as described above and immersed in glyoxylic acid solutions with varying proportions for more than 24 h. Finally, the hydrogels were dialyzed in ddH_2_O for 24 h to remove the unreacted glyoxylic acid. The modification of BSA-PEG hydrogels using bromoacetic acid and the modification of hemoglobin-PEG hydrogels using glyoxylic acid were achieved using the same methods. The preparation and modification of hemoglobin-PEG hydrogels were achieved using the same method as described above. The concentrations of hemoglobin and 4-armed PEG-SG were 1.5 and 5 mM, respectively. For the preparation of the hydrogels using modified protein, BSA was modified using glyoxylic acid as previously described before gelation.

Scanning electron microscope (SEM) imaging: SEM images were obtained using a Quanta Scanning Electron Microscope (Quata 200, FEI) at 20 kV. The hydrogels were lyophilized and coated with Pt prior to the measurement. More than 3 samples of each hydrogel were used in the SEM imaging and more than 4 images were randomly taken for each sample. The meshes were marked with ImageJ and the size was measured according to the image scale.

Thiol detection using 5,5′-dithiobis-(2-nitrobenzoic acid) (DTNB): The unfolding of BSA was characterized by detecting the exposed thiol in the proteins. Typically, hydrogels were prepared as described above and soaked in deionized water for 24 h. DTNB was then diluted into the solution, and the concentration of the reaction product of thiol and DTNB, 3-carboxy-4-nitrophenyl disulfide, was monitored by monitoring the UV absorbance at 412 nm.

Circular Dichroism Spectra (CD): Typically, modified BSA was suspended in ddH_2_O to a concentration of 1.5 μM. Then, CD spectra of all samples were recorded using a J-815 (JASCO Inc., Japan) spectrophotometer. The cuvette width was 1 cm, and the bandwidth was 0.2 nm.

Compressive test: The compressive stress–strain measurements were performed using a tensile-compressive tester (Instron-5944 with a 2 kN sensor) in air. In the compression–crack test, the rate of compression was kept constant at 20% min^−1^ with respect to the original height of the hydrogel, roughly in the range of 1.6–2.0 mm min^−1^. In the compression–relaxation cycle test, the rate of compression was also kept constant at 20% min^−1^ with respect to the original height of the hydrogel. The mechanical recovery percentage of each cycle was caudated as the highest stress in the current cycles divided by the highest stress in the first cycles. The fracture energy (*E_f_*) was measured based on the force-distance curves of the hydrogels that were compressed until fracture. The stress (*σ*) was calculated as the compression force divided by the cross-section area. *E_f_* was calculated by the integration of the area below the compression force-distance curves until the fracture point. The equation used in the calculation was as follows: $E_{f}=\int_{x_{0}}^{x_{f}}\sigma \left(x\right)dx$, in which *x*_0_ and *x_f_* correspond to the starting distance and the fracture distance of the compression, respectively. The Young’s modulus was the approximate linear fitting value of the stress–strain curve in the strain range of ~10%. The differential moduli were calculated as the slope of the stress–strain curve.

## 4. Conclusions

In summary, we demonstrated a universal method to tune the strain-stiffening property of protein-based hydrogels. By modifying the surface charge of the protein in BSA-PEG hydrogels, compressive strain-stiffening was significantly enhanced, while the bulk property, microstructure and recovery property remained unchanged. Due to the enhanced electrorepulsive interactions, the differential modulus of the modified BSA-PEG hydrogels increased by more than 3300% in the strain range of 0–60%, significantly higher than the ~1100% of the unmodified hydrogels. Furthermore, enhancements and weakening of strain stiffening were observed in BSA-PEG hydrogels modified using bromoacetic acid and hemoglobin-PEG hydrogels modified using glyoxylic acid, demonstrating the universal feasibility of charge modification on the regulation of the strain-stiffening property. We anticipate that this new method can be widely applied to independently tune the strain-stiffening performance of protein hydrogels without affecting the bulk properties, microstructures and biocompatibility.

## Figures and Tables

**Figure 1 ijms-23-03032-f001:**
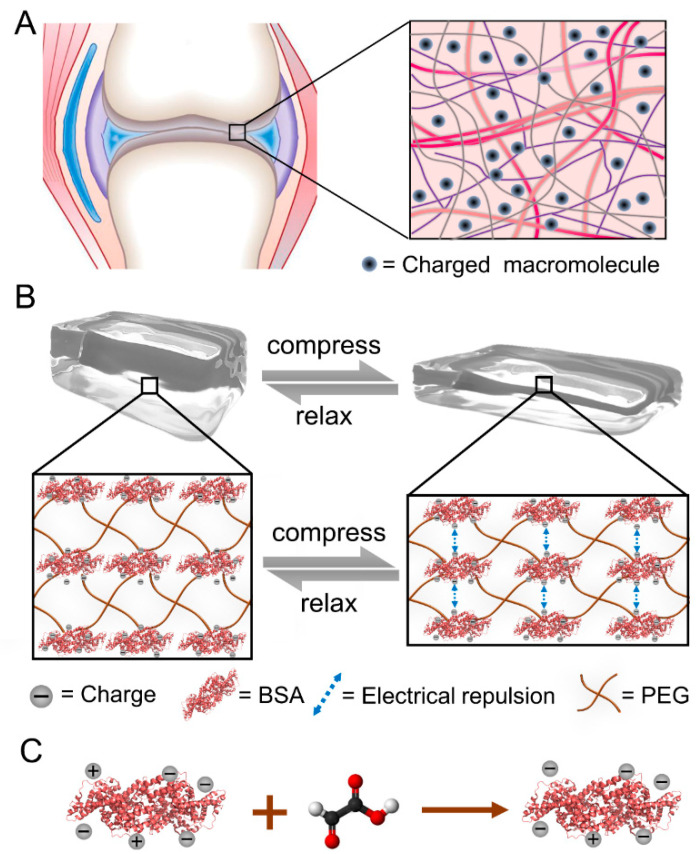
Illustration of tuning the strain-stiffening of BSA-PEG hydrogels by surface charge modification of proteins. (**A**) Schematic of cartilage with charged proteoglycans dispersed inside. (**B**) Compression–relaxation cycle of the BSA-PEG network and the electrical repulsion between adjacent BSA with charge modified under compression. (**C**) Surface charge modification of BSA using glyoxylic acid.

**Figure 2 ijms-23-03032-f002:**
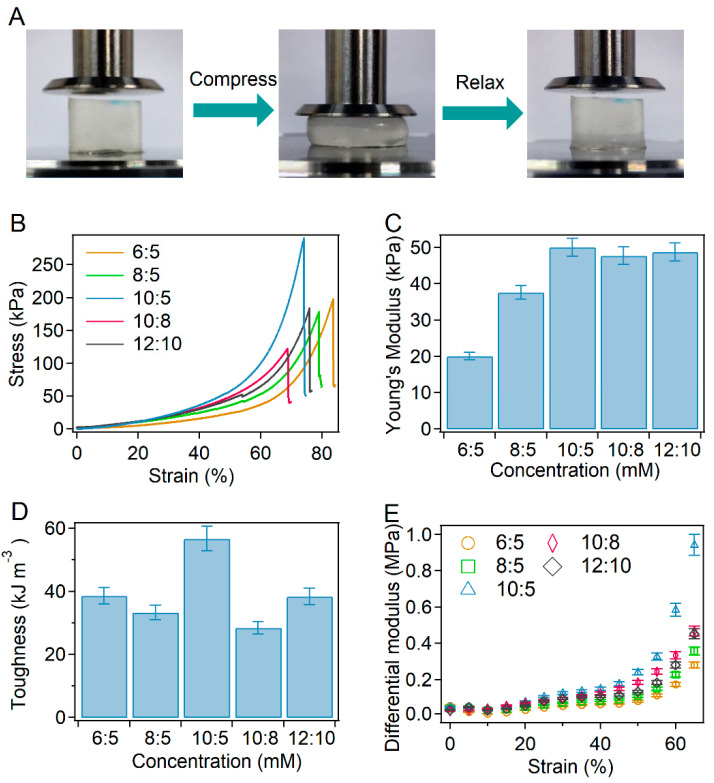
Compressive mechanical properties of BSA-PEG hydrogels. (**A**) Optical images of compression and relaxation of BSA-PEG hydrogels. (**B**) Typical stress–strain curves under compression for BSA-PEG hydrogels at different BSA:PEG ratios and solid contents in air. The concentrations of PEG and BSA were 6 and 5 mM, 8 and 5 mM, 10 and 5 mM, 10 and 8 mM, and 12 and 10 mM, respectively. (**C**,**D**) Young’s modulus (**C**) and toughness (**D**) correspond to BSA-PEG hydrogels at different BSA:PEG ratios and solid contents. (**E**) Differential modulus of BSA-PEG hydrogels at various strains.

**Figure 3 ijms-23-03032-f003:**
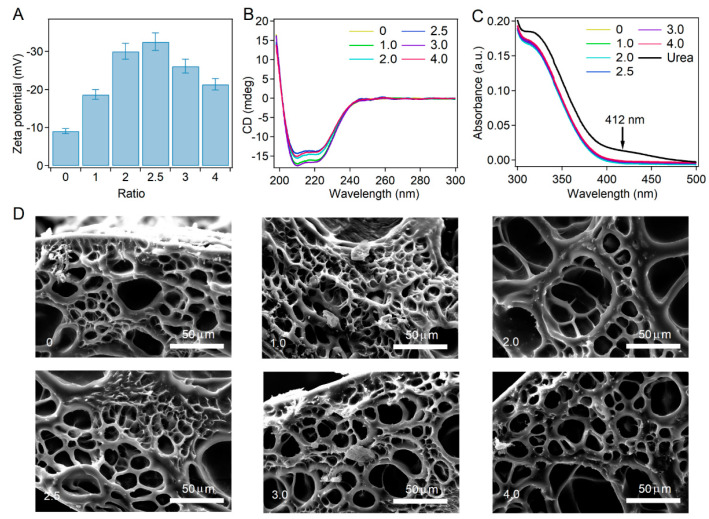
Surface charge modification of BSA and bulk properties of modified BSA-PEG hydrogels. (**A**,**B**) Zeta potential (**A**) and CD spectra (**B**) of modified BSA. Different molar ratios of glyoxylic acid and BSA were used in the modification (0:0, 1:1, 2:1, 2.5:1, 3:1 and 4:1, simplified as 0, 1.0, 2.0, 2.5, 3.0 and 4.0). (**C**) UV absorbance of DTNB-containing leachates of BSA-PEG hydrogels modified at different ratios of glyoxylic acid and BSA. The UV absorbance at 412 nm indicates the reaction products of DTNB, which were used to indicate the exposed thiol from unfolded BSA in hydrogels. (**D**) SEM images of BSA-PEG hydrogels modified at different ratios of glyoxylic acid and BSA.

**Figure 4 ijms-23-03032-f004:**
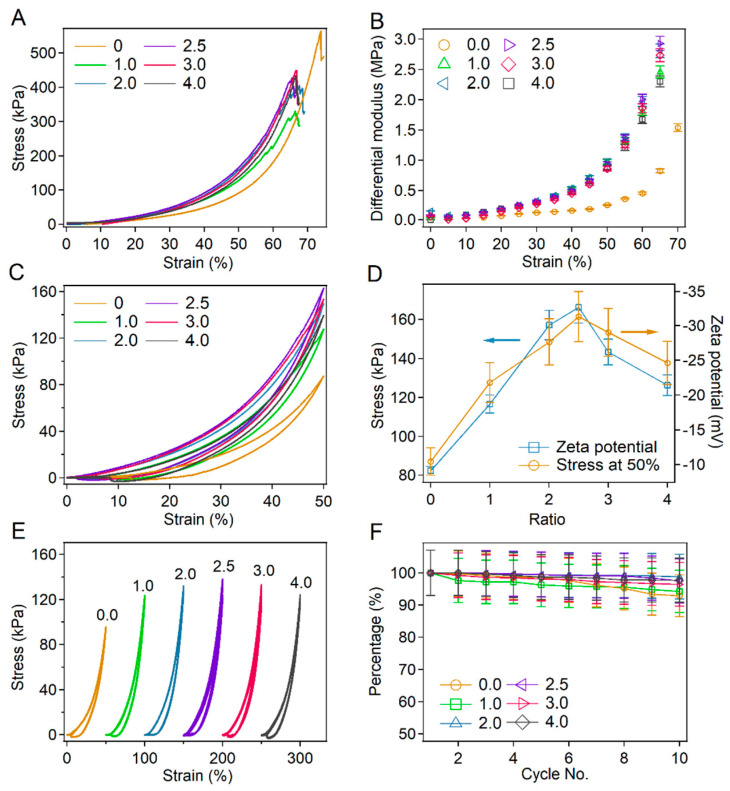
Compressibility of the BSA-PEG hydrogels modified with glyoxylic acid. (**A**) Compressibility of BSA-PEG hydrogels modified at different ratios of glyoxylic acid and BSA. Different molar ratios of glyoxylic acid and BSA were used in the modification (0:0, 1:1, 2:1, 2.5:1, 3:1 and 4:1, simplified as 0, 1.0, 2.0, 2.5, 3.0 and 4.0). (**B**) Differential modulus corresponding to BSA-PEG hydrogels modified at different ratios of glyoxylic acid and BSA. (**C**) Compression–relaxation of modified BSA-PEG hydrogels at the strain of 50%. (**D**) Summarized stress for modified BSA-PEG hydrogels at the strain of 50% and zeta potentials of BSA modified at different ratios of glyoxylic acid and BSA. (**E**) Continuous compression–relaxation cycles of BSA-PEG hydrogels modified with different ratios of glyoxylic acid and BSA for 10 cycles. (**F**) Normalized maximum stress of BSA-PEG hydrogels modified at different ratios of glyoxylic acid and BSA in 10 cycles of compression and relaxation.

**Figure 5 ijms-23-03032-f005:**
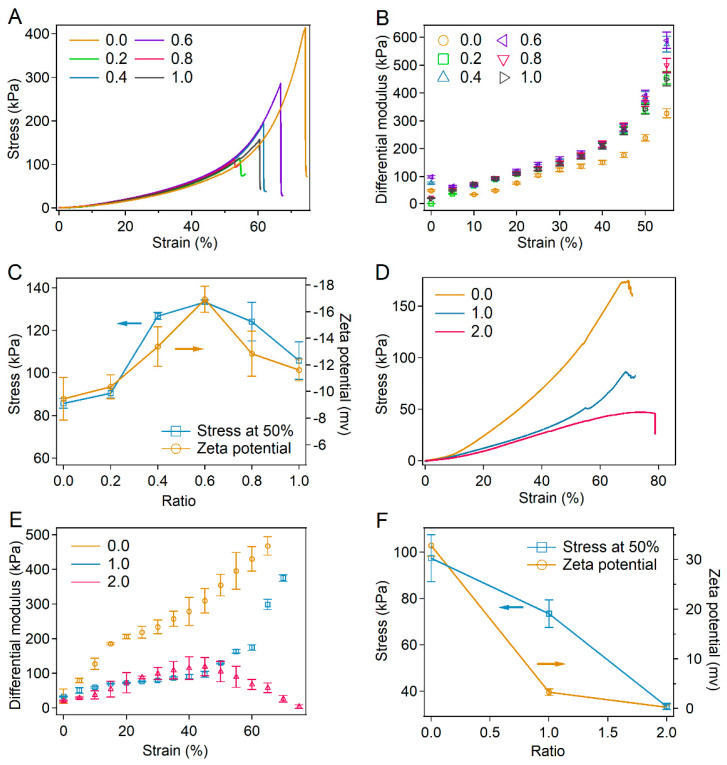
Mechanical properties of BSA-PEG hydrogels modified with bromoacetic acid and hemoglobin-PEG hydrogels modified with glyoxylic acid. (**A**) Compressibility of BSA-PEG hydrogels modified with bromoacetic acid. Different molar ratios of bromoacetic acid and BSA were used in the modification (0:0, 0.2:1, 0.4:1, 0.6:1, 0.8:1 and 1.0:1, simplified as 0, 0.2, 0.4, 0.6, 0.8 and 1.0). (**B**) Differential modulus corresponding to BSA-PEG hydrogels modified with different ratios of bromoacetic acid and BSA. (**C**) Summarized stress for modified BSA-PEG hydrogels at the strain of 50% and zeta potentials of BSA at different ratios of bromoacetic acid and BSA. (**D**) Compressibility of hemoglobin-PEG hydrogels modified with glyoxylic acid. Different molar ratios of glyoxylic acid: hemoglobin were used in the modification (0:0, 1:1 and 2:1, simplified as 0.0, 1.0 and 2.0). (**E**) Differential modulus corresponds to hemoglobin-PEG hydrogels modified at different ratios of glyoxylic acid and hemoglobin. (**F**) Summarized stress for modified hemoglobin-PEG hydrogels at the strain of 50% and zeta potentials of hemoglobin modified at different ratios of glyoxylic acid and hemoglobin.

## Data Availability

All data are available in the main text or the Appendix A.

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
