# Peer review of "Tuning Strain Stiffening of Protein Hydrogels by Charge Modification"

_ijms, 2022, doi:10.3390/ijms23063032_

Round 1

Reviewer 1 Report

In the response letter, the authors addressed all the points that I raised, and accordingly they also revised the content of manuscript. After revision, the manuscript has been significantly improved. Therefore, I recommend publication of this work in ijms.

Author Response

We thank the reviewer for the positive comments.

Reviewer 2 Report

This reviewer appreciates the authors’ careful consideration of all previous comments and additional experiments carried out to test their arguments. The authors have addressed most of my concerns. However, I do have one question about the new results added:

In the updated manuscript, the authors wrote, “The modified BSA-PEG hydrogels were split into particles to expose the inner surfaces and the zeta potentials were measured. As shown in Figure S3, the samples exhibited zeta potentials and pH values similar to those of the modified BSA, suggesting the successful surface charge modification of BSA inside hydrogels.” From the experimental results, the author found that BSA inside the hydrogel has been also modified. Thus, this technique should not be referred as “surface charge modification.” I would suggest the authors modify their terminology by taking out the “surface” term, as this is a global charge modification technique applied to the bulk sample.

Reviewer 3 Report

We thank the authors for considering most of our suggestions. However, there are still some questions that have not been addressed properly:

  1. Authors have calculated pore size (Figure S4C, Figure S11B and Figure S13C), but have not included the procedure followed to do so in the materials section.
  2. Since pH remains stable at all glyoxylic acid ratios (Figure S3), this seems not to be a coherent explanation for the the drop in z-potential at glyoxylic acid rations above 2.5 as speculated in lines 142-145. The authors may want to acknowledge they do not know why the change in Z-potential at high glyoxylic acid concentrations is happening and/or provide alternative explanations.
  3. In Figure S8 authors show a SDS-PAGE of BSA modified with different ratios of glyoxylic acid. However, there is not a clear BSA band at 66kDa, which makes it difficult to see whether the proportion of this band changes with the amount of glyoxylic acid (this may be solved by loading 5-10x less sample since the lanes appear to be overloaded). In the presented gel, there is a noticeable increase in intensity of the lowest mobility band, which may be suggestive of BSA crosslinking. Hence, the authors may want to acknowledge that a contribution of intermolecular crosslinking to their results (at least in the gels treated with glyoxylic acid) cannot be ruled out.
